# Encapsulated Allografts Preclude Host Sensitization and Promote Ovarian Endocrine Function in Ovariectomized Young Rhesus Monkeys and Sensitized Mice

**DOI:** 10.3390/bioengineering10050550

**Published:** 2023-05-03

**Authors:** James R. Day, Colleen L. Flanagan, Anu David, Dennis J. Hartigan-O’Connor, Mayara Garcia de Mattos Barbosa, Michele L. Martinez, Charles Lee, Jenna Barnes, Evan Farkash, Mary Zelinski, Alice Tarantal, Marilia Cascalho, Ariella Shikanov

**Affiliations:** 1Department of Biomedical Engineering, University of Michigan, Ann Arbor, MI 48109, USA; 2Department of Medical Microbiology and Immunology, University of California, Davis, CA 95616, USA; 3California National Primate Research Center, University of California, Davis, CA 95616, USA; 4Department of Surgery, University of Michigan, Ann Arbor, MI 48109, USA; 5Department of Pediatrics, University of California, Davis, CA 95616, USA; 6Department of Cell Biology and Human Anatomy, University of California, Davis, CA 95616, USA; 7Department of Pathology, University of Michigan, Ann Arbor, MI 48109, USA; 8Division of Reproductive & Developmental Sciences, Oregon National Primate Research Center, Beaverton, OR 97006, USA; 9Department of Obstetrics & Gynecology, Oregon Health & Science University, Portland, OR 97239, USA; 10Department of Microbiology & Immunology, University of Michigan, Ann Arbor, MI 48109, USA; 11Department of Obstetrics and Gynecology, University of Michigan, Ann Arbor, MI 48109, USA; 12Department of Macromolecular Science & Engineering, University of Michigan, Ann Arbor, MI 48109, USA; 13Cellular and Molecular Biology, University of Michigan, Ann Arbor, MI 48109, USA

**Keywords:** ovarian endocrine function, immunoisolation, hydrogel

## Abstract

Transplantation of allogeneic donor ovarian tissue holds great potential for female cancer survivors who often experience premature ovarian insufficiency. To avoid complications associated with immune suppression and to protect transplanted ovarian allografts from immune-mediated injury, we have developed an immunoisolating hydrogel-based capsule that supports the function of ovarian allografts without triggering an immune response. Encapsulated ovarian allografts implanted in naïve ovariectomized BALB/c mice responded to the circulating gonadotropins and maintained function for 4 months, as evident by regular estrous cycles and the presence of antral follicles in the retrieved grafts. In contrast to non-encapsulated controls, repeated implantations of encapsulated mouse ovarian allografts did not sensitize naïve BALB/c mice, which was confirmed with undetectable levels of alloantibodies. Further, encapsulated allografts implanted in hosts previously sensitized by the implantation of non-encapsulated allografts restored estrous cycles similarly to our results in naïve recipients. Next, we tested the translational potential and efficiency of the immune-isolating capsule in a rhesus monkey model by implanting encapsulated ovarian auto- and allografts in young ovariectomized animals. The encapsulated ovarian grafts survived and restored basal levels of urinary estrone conjugate and pregnanediol 3-glucuronide during the 4- and 5-month observation periods. We demonstrate, for the first time, that encapsulated ovarian allografts functioned for months in young rhesus monkeys and sensitized mice, while the immunoisolating capsule prevented sensitization and protected the allograft from rejection.

## 1. Introduction

Over the past few decades, childhood cancer survival rates have increased substantially, reaching over 85% due to advances in new therapies. However, the same cancer therapies can potentially lead to a myriad of health complications in childhood cancer survivors [1,2,3,4,5]. One of the most common complications for female cancer survivors is premature ovarian insufficiency (POI), caused by gonadotoxic treatments [6,7]. POI leads to depletion of the follicular pool, resulting in infertility and disruption of ovarian endocrine function. POI is particularly detrimental in children with a cancer diagnosis prior to puberty, where hormonal loss can impact the maintenance of homeostasis between the ovaries and other tissues and endocrine organs [8,9,10,11,12]. Changes during puberty also promote physical and psychological development into adulthood and can determine height, bone health, insulin responsiveness, lipid metabolism, cardiovascular health, and cognition [5,13]. These changes are orchestrated by the pulsatile secretion of gonadotropin-releasing hormone (GnRH) from the hypothalamus, which regulates the release of gonadotropins, luteinizing (LH), and follicle-stimulating (FSH) hormones from the pituitary. FSH and LH stimulate the production of ovarian hormones and peptides, including estradiol, androstenedione, progesterone, inhibins A and B, activin, and follistatin. The production of ovarian hormones is tightly regulated by a negative feedback loop that inhibits the production of GnRH and FSH in the brain and pituitary [14,15], known as the hypothalamic-pituitary-gonadal (HPG) axis, which is crucial for the development of the reproductive, musculoskeletal, cardiovascular, and immune systems as well as endocrine regulation [16,17]. In young patients with POI, the HPG axis is disrupted due to deficiencies in ovarian hormones, which leads to imbalances across the endocrine system, resulting in comorbidities such as suboptimal bone development, metabolic changes, and abnormal fat deposition [18,19,20].

Currently, the only clinically available option to treat POI in adolescent girls to initiate puberty is hormone replacement therapy (HRT), which delivers gradually increasing, yet fixed, amounts of estrogen [20,21]. Hormone replacement therapy was originally designed to treat postmenopausal symptoms in women, and thus the impact on children is not well-established [17,21]. Additionally, HRT delivers only a fraction of ovarian hormones in a non-pulsatile manner, which does not mimic physiological puberty or the complexity of the HPG axis. In turn, non-physiologic HRT given to induce puberty in girls with POI can also lead to premature closure of growth plates, cessation of bone growth, and metabolic imbalances [22,23]. Auto-transplantation of cryopreserved ovarian tissue banked prior to anti-cancer treatments is a potential option under investigation. Transplanted autologous ovarian tissue restored endocrine function and the transition to puberty in girls [7,11,24] and has led to more than 100 babies born to either identical twin sisters or cancer survivors who cryopreserved their ovaries [25,26,27,28,29]. However, this option is associated with the risk of reintroducing malignant cells potentially harbored within the ovary, particularly in patients with hematological malignancies, the most common childhood cancer [30,31,32]. Although malignant cells can be detected in some cancers, there has been no universally safe protocol established to screen ovarian tissue, contributing to an uncertain risk of re-seeding in those undergoing autologous transplantation. In addition, most childhood cancer patients still do not cryopreserve tissue prior to treatment [10,33,34]. 

We hypothesized that donor allogeneic ovarian tissue implanted subcutaneously would respond to circulating gonadotropins and secrete ovarian hormones, mimicking the physiological process while mitigating the risk of re-introduction of cancer cells. Previously, we developed and optimized a novel, dual-layered, poly(ethylene glycol)-based (PEG) capsule that sustains the survival and function of allogeneic mouse ovarian tissue while preventing rejection and supporting ovarian development in immune competent mice [35,36,37,38,39,40]. The capsule consists of a degradable hydrogel core that provides a supportive environment for the implanted tissue, promoting the cyclical changes associated with folliculogenesis, surrounded by a non-degradable hydrogel shell that allows free diffusion of essential metabolites and nutrients while preventing rejection by blocking the infiltration of host immune cells. Our previous studies in mice have demonstrated that encapsulated ovarian allografts undergo folliculogenesis, producing ovarian hormones in a regulated physiological manner, restoring endocrine function in a mouse model of POI without sensitizing the immune system of the host [35,38,39].

In a clinical setting, each patient may require recurring allo-transplantations due to the finite number of follicles transplanted in each graft. Recurring exposure to allografts may sensitize the host and, as a result, accelerate rejection and shorten the duration of graft function. The goal of this study was to test whether encapsulated murine ovarian allografts could function in previously sensitized hosts and whether this technology translates to larger animals, such as Rhesus macaques. We demonstrated that the capsule protected the ovarian allograft in a pre-sensitized murine host that otherwise rejects non-encapsulated allografts within days of transplantation [41,42]. Second, we implanted encapsulated auto- and allografts in young ovariectomized monkeys, which are anatomically and physiologically closer to humans, towards the translation of this approach [43,44,45,46,47,48]. Our findings suggest that the auto- and allogeneic ovarian tissues implanted in ovariectomized mice and rhesus monkeys survived in the capsule, secreted estradiol and progesterone, and did not elicit immune responses or rejection. We demonstrate, for the first time, that encapsulated ovarian allografts functioned for months while the immunoisolating capsule prevented sensitization and protected the allograft from rejection.

## 2. Materials and Methods

### 2.1. Animals

All animal procedures conformed to the requirements of the Animal Welfare Act, and protocols were approved prior to implementation by the Institutional Animal Care and Use Committee (IACUC) at the University of Michigan for the rodents (PRO00007716) and the University of California, Davis for rhesus monkeys (see below).

Activities related to nonhuman primate animal care (diet, housing) were performed per IACUC-approved California National Primate Research Center standard operating procedures (SOPs). Physical signs were monitored daily, and body weights were assessed each time the animals were sedated with ketamine (10 mg/kg).

### 2.2. Study Design

#### 2.2.1. Murine Model

Adult female mice (BALB/c) underwent bilateral ovariectomies to induce POI. As a positive control for sensitization, BALB/c mice were implanted subcutaneously with a non-encapsulated ovary from 6- to 8-day-old CBA × C57BL/6 (F1) mice for 42 days (Figure 1A) (*n* = 7), followed by a subsequent implantation of another non-encapsulated F1 ovary for 14 days in a subset of these mice (*n* = 4 mice). For experimental groups, BALB/c mice were either (i) implanted with encapsulated ovarian allogeneic (F1) tissue for 60 days followed by a subsequent implantation of encapsulated ovarian allografts for another 60 days (*n* = 5 mice) or (ii) implanted with non-encapsulated F1 ovarian tissue for 60 days to induce sensitization followed by a subsequent implantation of encapsulated ovarian allografts for 60 days (*n* = 3 mice) (Figure 1D). A total of 15 mice were used in these studies. Each mouse was implanted with two replicate grafts or non-encapsulated grafts.

Bilateral ovariectomies were performed on adult female mice (BALB/c) at 12–16 weeks of age. The mice were anesthetized with isoflurane for the surgical procedure, and Carprofen (5 mg/kg body weight, Rimadyl, Zoetis, Kirkland, Quebec, Canada) was administered subcutaneously for analgesia. The intraperitoneal space was exposed through a midline incision in the abdominal wall, secured using an abdominal retractor. The ovaries were removed, and the muscle and skin layer of the abdominal wall were closed with 5/0 absorbable sutures (AD Surgical). The mice recovered in a clean, warmed cage and received another dose of Carprofen 12 h post-recovery or as needed.

#### 2.2.2. Collection of Donor Murine Ovaries

Ovaries from 6- to 8-day-old CBA × C57BL/6 (F1) mice were collected and transferred to Leibovitz L-15 media (P/N 11415, Gibco, Saint Louis, MO, USA). The ovaries were dissected into 2–4 pieces and transferred into maintenance media (α-MEM, P/N 32561, Gibco, Saint Louis, MO, USA) and placed in a CO_2_ incubator prior to encapsulation.

#### 2.2.3. Hydrogel Preparation and Murine Ovarian Tissue Encapsulation

The degradable core of the PEG-based capsule was prepared with 8-arm PEG-VinylSulfone (PEG-VS) (40kDa, Jenkem Technology, Beijing, China) and cross-linked with a plasmin-sensitive tri-functional peptide sequence (Ac-GCYK↓NSGCYK↓NSCG, MW 1525.69 g/mol, >90% Purity, Genscript, ↓ indicates the cleavage site of the peptide). The non-degradable shell of the capsule was prepared using 4-arm PEG-VS (20 kDa, Jenkem Technology, Beijing, China) with Irgacure 2959 (BASF, Switzerland, MW = 224.3) and 0.1% N-vinyl-2-pyrrolidone (P/N V3409, Sigma-Aldrich, St. Louis, MO, USA). The detailed protocol is described in Day et al. [33,37]. The murine ovarian fragments were first encapsulated in 4 µL of the degradable PEG hydrogel for 5 min. The degradable core and graft were then placed in the center of a 10 µL droplet of PEG-VS precursor solution (5% *w*/*v* PEG-VS, 0.4% Irgacure 2959, 0.1% NVP) and exposed to UV light for 6 min. All constructs were imaged (with a Leica M60 stereo microscope) immediately after encapsulation of the tissue to confirm encapsulation.

#### 2.2.4. Subcutaneous Implantation in Mice

A small incision was made on the dorsal side of anesthetized mice (BALB/c), and the immunoisolating capsules with the ovarian allografts or non-encapsulated ovarian allografts were implanted subcutaneously. The skin was closed using 5/0 absorbable sutures. The mice received Carprofen for analgesia for at least 24 h after surgery or as needed.

#### 2.2.5. Vaginal Cytology in Mice

Daily vaginal cytology was performed after ovariectomy to assess estrous cycle status and confirm cessation for seven days post-surgery. Starting one week after allograft implantation, vaginal cytology was performed daily to assess estrus cycle status and determine if estrous cyclicity resumed until euthanasia. Observation of the transition from leukocytes to cornified cells at least once a week was the criteria used to determine a resumed or continued cycle.

#### 2.2.6. Flow Cytometry of Mouse Serum

Serum alloantibody titer measurements were performed using flow cytometry before and after implantation. They were reported as MFI for the highest dilution showing fluorescence detectable above background (non-immune serum from a non-implanted mouse) in immunized mice (positive controls implanted with non-encapsulated ovary allografts). Thymocytes were isolated from CBA × C57BL/6 donor mice and incubated with serially diluted recipient serum for 30 min at 4 °C. Antibodies bound to the thymocytes were detected by Cy5-conjugated goat anti-mouse IgG (1:250 dilution, 1030-15, Southern Biotech, Birmingham, AL, USA) and Alexa Fluor 488-conjugated goat anti-mouse IgM (1:250 dilution, 1020-30, Southern Biotech, Birmingham, AL, USA) for 30 min at 4 °C and analyzed in a BD FACSCanto II (BD Biosciences, Franklin Lakes, NJ, USA). The MFI in the APC-channel (measuring bound IgG) and FITC channel (measuring bound IgM) were determined with FlowJo 10 software (FlowJo, LLC, Ashland, OR, USA).

#### 2.2.7. Histological Analysis of the Murine Ovarian Allografts and Encapsulated Ovarian Allografts

Following euthanasia, the immunoisolating capsules or allografts were retrieved from the animals, fixed in 4% paraformaldehyde at 4 °C overnight, then transferred and stored in 70% ethanol at 4 °C. Samples were histologically processed for paraffin embedding, serially sectioned at 5 μm thickness, and stained with hematoxylin and eosin (H&E).

#### 2.2.8. Immunohistochemistry (IHC) of Mouse Ovarian Allografts

To analyze T cell infiltration into non-encapsulated or encapsulated ovarian allografts, paraffin-stained sections were used to identify CD8^+^ T cells. Following deparaffinization with xylene and rehydration, the sections were incubated in antigen retrieval buffer, pH 9.0 (ab94681, Abcam, Cambridge, MA, USA), for 20 min at 97 °C and an additional 20 min at room temperature to cool. Next, the slides were incubated with KPL Universal Block (5560-0009, SeraCare Life Sciences, Milford, MA, USA) to block non-specific binding sites for 30 min at room temperature. The sections were incubated at room temperature for 1 h with the primary rabbit polyclonal anti-mouse CD8 antibody (1:500 dilution, ab203035, Abcam, Cambridge, MA, USA). The slides were subsequently incubated at room temperature with a secondary goat anti-rabbit antibody (1:50 dilution for 30 min, 4010-05, Southern Biotech, Birmingham, AL, USA). Diaminobenzidine (DAB) (BDB2004L, Betazoid DAB Chromogen Kit, BioCare Medical, Pacheco, CA, USA) was used as a chromogen for 10 min at room temperature. Hematoxylin (220-102, Fischer Scientific, Kalamazoo, MI, USA) was used as a counterstain. For negative controls, paraffin sections were incubated without the primary antibody. To assess the presence or absence of CD8^+^ T cells, 12 sections from the front, middle, and end of each specimen were examined to represent the full thickness of the implant. Five equal-sized fields in the four corners and center of each section were assessed for positive DAB chromogen staining.

### 2.3. Rhesus Monkey Model

#### 2.3.1. Ovariectomies and Subcutaneous Implantation in Recipient Rhesus Monkeys

Rhesus monkeys (~3 years of age, ~4 kg) were sedated with ketamine (10 mg/kg; IM) and prepared for bilateral ovariectomy according to established protocols [49]. Briefly, atropine was given (0.04 mg/kg), followed by intubation for the administration of isoflurane (to effect), and an indwelling catheter was placed for intravenous (IV) fluids. A small midline incision was made, and the ovaries were exposed, removed individually, and placed in a sterile culture dish for processing in a biosafety cabinet under aseptic conditions.

Ovarian tissue from Primate A was encapsulated in the immunoisolating hydrogel under aseptic conditions and implanted in both Primate A (an autograft) and Primate B (an allograft). Once the encapsulated grafts were ready for implantation, a small incision (~0.5 cm) was made between the scapulae. The encapsulated tissue was gently inserted under aseptic conditions, and the incisions were sutured closed and reinforced with skin glue. Blood and urine samples were collected regularly, as described below. All animals were monitored post-surgery per SOP and administered analgesics post-operatively.

At approximately 5 months post initial implantation surgery, animals were prepared for a second implantation surgery. Primate C was ovariectomized and received fresh encapsulated autologous tissue grafts (right ovary) as described above. The first implantation round of encapsulated ovarian grafts was performed on Primates A and B, which then received encapsulated allogeneic tissue from Primate C. At approximately four months post-implantation, the encapsulated tissue was collected for analysis from all three animals.

Thus, overall, two rounds of subcutaneous implantations of encapsulated autologous and allogeneic ovarian tissue fragments were performed. In the first round, one animal received 20 encapsulated autologous ovarian fragments, and 1 animal received 20 encapsulated allogeneic ovarian fragments. In the second round, three animals (two animals from the first round and an additional animal C) received 20 encapsulated ovarian fragments (N = 1 autologous and N = 2 allogeneic).

#### 2.3.2. Primate Ovarian Tissue Encapsulation

The primate ovarian allograft tissue was encapsulated using the same capsule formulation as described above for the murine model. The primate ovaries were dissected into cubes measuring 1 × 1 × 1 mm (1 mm^3^) contributing to approximately 60 fragments, which were transferred into maintenance media (α-MEM, P/N 32561, Gibco, Saint Louis, MO, USA) prior to encapsulation. The monkey ovarian tissue fragments were first individually encapsulated in an 8 µL degradable PEG core, allowed to cross-link for 5 min, and then placed in the center of a 20 µL droplet of PEG-VS precursor solution (5% *w*/*v* PEG-VS, 0.4% Irgacure 2959, and 0.1% NVP) and exposed to UV light for 6 min.

#### 2.3.3. Primate Urinary Estrone Conjugate (E1C) and Pregnanediol Glucuronide (PdG) Analysis

Urine samples were collected daily from cage pans according to SOP, starting one month prior to surgery and through most of the post-implantation period [49]. Rhesus monkeys were housed to ensure urine was collected from individual animals. Collected samples were centrifuged at 1500 rpm for 10 min, and the supernatant (~2–3 mL) was transferred to a cryogenic vial (Caplugs Evergreen, Buffalo, NY, USA) and stored at ≤−20 °C until processed. Urinary E1C and PdG levels were determined by the Endocrine Core (University of California, Davis) via the analysis of the urine supernatant according to established protocols [21].

#### 2.3.4. Mixed Primate Lymphocyte Culture

PBMCs were isolated from blood samples collected monthly by centrifugation onto a step gradient of Lymphocyte Separation Medium (LSM; MP Biomedicals, LLC, Solon, OH, USA). Stimulator cells were treated with 40 µg/mL mitomycin C (Sigma) at 37 °C for 30 min. CFSE-labeled responder cells were stimulated with an equal number of unlabeled stimulator cells or with plate-bound anti-CD3 antibodies (positive control; clone SP34-2). Cells were harvested after 4–6 days of incubation, stained with fluorescently labeled antibodies, including those specific for CD4 and CD8 (clones L200 and 3B5, respectively); cytometry data were collected on a BD Fortessa; and the results were analyzed in FlowJo’s proliferation platform (FlowJo, LLC, Ashland, OR, USA).

#### 2.3.5. Histological Analysis of the Encapsulated Ovarian Primate Allografts

Following euthanasia, the immunoisolating capsules with ovarian tissue grafts were retrieved from the animals, fixed in 4% paraformaldehyde at 4 °C overnight, transferred, and stored in 70% ethanol at 4 °C. During the encapsulation process of the ovarian grafts, fresh non-encapsulated ovarian tissue samples from each donor were also fixed and stored similarly. Samples were histologically processed for paraffin embedding, serially sectioned at 5 μm thickness, and stained with H&E.

### 2.4. Statistics

Statistical analysis was performed using GraphPad Prism software version 9.0.0. For the murine animal model IgG MFI measurements, multiple comparisons were made using a repeated measures one-way ANOVA test with the Geisser–Greenhouse correction. Dunnett’s multiple comparison test was used to determine which timepoint groups were significantly different from the control group (pre-implantation). The data passed the Shapiro–Wilk test, which was used to assess the normal distribution of data in this dataset. For the murine animal model CD8+ cell measurements, multiple comparisons were made using a one-way ANOVA test with an assumed Gaussian distribution, supported by the Shapiro–Wilk test. Tukey’s multiple comparison test, with a single pooled variance, was used to determine which groups were significantly different from one another. The ANOVA tests are one-tailed. For the primate animal model mixed lymphocyte culture CD4+ and CD8+ cell measurements, a repeated measures ANOVA (main effects only) was conducted to determine the effect of treatment (allograft vs. control). The data passed the Shapiro–Wilk test, which was used to assess the normal distribution of data in this dataset.

## 3. Results

This section may be divided into subheadings. It should provide a concise and precise description of the experimental results, their interpretation, and the experimental conclusions that can be drawn.

### 3.1. Immunoisolating Capsule Prevents Sensitization of the Host: Studies in a Murine Model

Sensitization of the host and the presence of circulating alloantibodies may shorten graft longevity and elicit local and systemic adverse effects. Here, our objective was to demonstrate that encapsulation of ovarian allograft tissue prevented sensitization of the host, which would allow repeated implantations of encapsulated allografts without the risk of rejection. As expected, implantation of non-encapsulated allogeneic ovarian tissue elicited an immune response mediated by T and B cells, resulting in sensitization of the host against the donor with a 60-fold increase in allo-specific IgG and cell-mediated destruction of the graft tissue (Figure 1A—study design, Figure 1B,C). Allo-specific IgG antibodies significantly increased from undetectable post-implantation to an average of 1123 mean fluorescence intensity (MFI) on day 28 (*, *p* = 0.0036) and to an average of 1256 MFI on day 42 (*, *p* = 0.0046) (*n* = 7), confirming the allogeneic immune reaction between the two strains of mice. We then assessed if grafting of allogeneic ovarian tissue encapsulated in a dual-layered immunoisolating capsule allowed repeated implantation without the risk of rejection (Figure 1D—study design). The immune-isolating capsule creates a barrier between the allograft and the host to minimize the host’s exposure to alloantigens and prevent stimulation of alloimmunity and graft damage owing to pre-existent alloimmunity. In contrast to non-encapsulated controls, encapsulated ovarian allografts did not sensitize recipients, which was confirmed by undetectable levels of circulating alloantibodies. Following the first and subsequent implantations of encapsulated ovarian allografts (Figure 1D), the levels of circulating allo-specific IgGs remained at the pre-implant level, ranging from non-detectable to 138 MFI for up to 60 days post-implantation (Figure 1E,F). It should be noted that there was a statistically significant difference between IgG MFI values pre-implantation (−1 day timepoint) and 60 days after the first round of implantation, as well as 7 and 28 days after the second round of implantation; however, the small difference in IgG MFI values detected is not biologically significant. We concluded that neither primary nor secondary implantation of encapsulated allografts sensitized recipients because the levels of circulating IgG alloantibodies did not increase after the implantations of encapsulated allografts.

### 3.2. Immunoisolating Capsule Prevents Rejection of Allogeneic Ovarian Tissue and Supports Endocrine Function in a Murine Model

Graft morphology and function were compared between mice implanted with non-encapsulated (Figure 2A) and encapsulated allografts (Figure 2D–F). The non-encapsulated ovarian allografts were resorbed based on gross examination and histology. Histological analysis of the tissue retrieved from the implantation site demonstrated necrosis, the absence of surviving healthy follicles, and the loss of distinctive ovarian morphology (Figure 2A). All ovariectomized mice implanted with non-encapsulated ovarian allografts resumed estrous cyclicity 1 week post-implantation, but the cyclicity ceased after 4 weeks and remained as persistent diestrus consistent with briefly restored yet failed ovarian function shortly thereafter (Figure 2B is a representative plot of estrous activity in an individual mouse, and Figure 2C shows the data combined for all mice in the control group). Allograft failure and loss of estrous activity were accompanied by an increase in circulating IgG alloantibodies (Figure 1B,C). In contrast, all ovariectomized mice implanted with encapsulated ovarian allografts resumed cyclicity 2 weeks after implantation, with 80% of the mice resuming cyclicity after 1 week of implantation. The cyclicity persisted for 60 days post-implantation when the grafts were retrieved (Figure 2G,H). Histological analysis of the allografts revealed healthy ovarian tissue with multiple follicles at different developmental stages (Figure 2D,E). Although our results suggest that the encapsulated ovarian tissue did not sensitize the host after one implantation, in a more stringent test we asked if a second allograft from the same donor was compatible with function. To address this question, we first implanted mice with encapsulated ovarian allografts for 60 days, followed by a second encapsulated ovarian allograft for an additional 60 days. Multiple healthy, developing follicles up to the antral stage (Figure 2F) were present in the allografts retrieved from the second implant, confirming that folliculogenesis was not affected by the first implant. All mice resumed estrous activity by 2 weeks following implantation and continued cycling through the second period of 60 days post-implantation, suggesting restoration of ovarian endocrine function. Because encapsulated allografts in the second round functioned as well as primary encapsulated allografts, we concluded that encapsulation may allow multiple implants without risking loss of function due to an immune response.

### 3.3. Encapsulated Allografts Restored Ovarian Endocrine Function and Were Shielded from Rejection in Prior Sensitized Murine Hosts

We demonstrated that encapsulated ovarian allografts implanted in naïve mice were protected from immune rejection even after repeated implantations. However, the question remained whether the encapsulated ovarian allografts were protected in a host that had been previously sensitized (Figure 3A). To answer this question, we first implanted a non-encapsulated ovarian allograft, which sensitized the host and caused the production of alloantibodies. The sensitization was confirmed by elevated levels of circulating alloantibodies (on average >1000 MFI, Figure 1B and Figure 3F) and by the cessation of estrous cycles approximately 3 weeks post-implantation (Figure 3D,E). After sensitization was confirmed, we implanted encapsulated ovarian allografts in the sensitized hosts. All sensitized recipients of encapsulated allografts resumed estrous cyclicity by 2 weeks post-implantation and continued cycling throughout the entire implantation period (Figure 3D,E). In spite of the presence of detectable circulating alloantibodies persisting from prior implantation of non-encapsulated ovarian tissue, histological analysis of the retrieved encapsulated ovarian allografts 60 days post-implantation revealed fully encapsulated ovarian tissue with multiple healthy, developed follicles at preantral and antral stages, similar to observations in encapsulated ovarian tissue implanted in naïve mice (Figure 3B,C).

Naïve animals implanted with non-encapsulated ovarian allografts maintained elevated circulating allo-specific IgG indicative of sensitization, 1100 MFI on average, by 28 days post-implantation. As a comparison, implantation of non-encapsulated ovarian tissue in sensitized recipients (*n* = 4) showed continued immune response, confirmed by a significant elevation in allo-specific IgG antibodies relative to naïve levels prior to implantation at 42 days post-implantation 1 (*p* = 0.0428) and 7 days post-implantation 2 (*p* = 0.0435) (Figure 3F). Immunohistochemical analysis of the non-encapsulated allografts (ovary) demonstrated significant infiltration of CD8^+^ T cells, consistent with rejection (Figure 3G,I).

In contrast, encapsulated allograft from the first (Capsule 1) and second (Capsule 2) implantations and the implantation of an encapsulated allograft following sensitization (S-Capsule) had no CD8^+^ cell infiltrate inside the graft (Figure 3G,H). Figure 3H is a representative histological image of the immunohistochemical (IHC) staining for CD8^+^ that was typical of all encapsulated allografts in these groups, illustrating the lack of CD8^+^ T cells within the allograft. Importantly, following the secondary implantation of an encapsulated allograft in a sensitized host (S-Capsule), the encapsulated allografts had no CD8^+^ T lymphocyte infiltration (Figure 3G,H), in spite of high levels of circulating allo-specific IgG indicative of sensitization. These observations suggest that capsules prevented infiltration of T cells and rejection of the allografts despite immune pre-sensitization.

### 3.4. Encapsulation and Implantation of Nonhuman Primate Ovarian Tissue in Ovariectomized Adolescent Rhesus Monkeys

We performed two rounds of subcutaneous implantations of encapsulated autologous and allogeneic ovarian tissue fragments in ovariectomized young rhesus monkeys (Figure 4). Follicles in the encapsulated and implanted ovarian autografts survived and resumed folliculogenesis and steroidogenesis. Consistent with the ovaries of peripubertal rhesus monkeys, the ovaries removed from the animals at the time of ovariectomy (Primates A and C) contained multiple primordial and primary follicles (Figure 5A–D). The cortical stroma had densely packed primordial follicles with oocytes surrounded by a single layer of flat squamous granulosa cells (Figure 5(Bi,Di)). In addition to primordial follicles, primary and a few small preantral follicles were also identified in the ovarian tissue. Primary follicles showed the characteristic oocyte surrounded by a single layer of cuboidal granulosa cells (Figure 5Bii,5Dii), while multilayered secondary follicles (preantral follicle) had a few layers of granulosa cells surrounding the oocyte (Figure 5(Biii–iv,Diii–iv)). As anticipated in adolescent animals, antral follicles were not observed in the cortical fragments, and the implanted fragments contained only primordial and primary follicles. After 4–5 months post-implantation, the capsules with autologous tissue were removed, fixed, and analyzed for follicular development by histological analysis. All stages of follicles, ranging from primordial to antral, were observed in the capsules containing autografts (Figure 5E–I). Importantly, the presence of healthy-appearing and growing follicles supported continued folliculogenesis and was consistent with measured basal levels of ovarian hormones.

We evaluated restoration of ovarian endocrine function by measuring urinary levels of E_1_C and PdG daily (no urine was not collected in Primates A and B between days 55 and 100, and Primate C between days 20 and 27 due to technical challenges). E_1_C and PdG levels were normalized to urinary creatinine (Cr) levels, as previously reported [50,51]. Before ovariectomy, all animals exhibited basal ovarian activity, with levels of E_1_C and PdG fluctuating between 10 and 51 ng/mg Cr for E_1_C and 14 and 63 ng/mg Cr for PdG, respectively (Figure 5J,K and Figure 6B, before Day 0). After ovariectomy followed by immediate implantation of autologous ovarian tissue encapsulated in immunoisolating capsules, the levels of E_1_C and PdG were 10–30 ng/mg Cr. In Primate A (Figure 5J), the levels of E_1_C increased 25 days post-implantation, reaching 50–70 ng/mg Cr. PdG also increased, reaching 55 ng/mg Cr, similar to the measured levels of E_1_C and PdG before ovariectomy. Primate C with autologous implants showed lower levels of circulating E_1_C and PdG before ovariectomy (Figure 5K), ranging between 10 and 28 ng/mg Cr for E_1_C and 20 and 40 ng/mg Cr for PdG, respectively. After ovariectomy and immediate implantation of encapsulated ovarian autografts, the levels of E_1_C and PdG remained in the same range for 35 days, then fluctuated, reaching peak values of 40 ng/mg E_1_C and 65 ng/mg PdG. In both animals with encapsulated ovarian autografts, the levels of E_1_C and PdG fluctuated from 6 to 69 ng/mg and 8 to 65 ng/mg Cr (Figure 5J,K), respectively, suggesting the encapsulated tissue was producing ovarian hormones at a basal level. A decline and peak occurring approximately every 7 days was considered typical for fluctuations of basal steroid levels.

### 3.5. Implanted Encapsulated Ovarian Allografts Secrete E_1_C and PdG and Elicit Minimal Immune Responses

Two adolescent female rhesus monkeys, Primate A and B, were implanted with encapsulated ovarian allografts. Before the ovariectomy they exhibited basal ovarian activity with fluctuating levels of urinary E1C and PdG between 15 and 65 ng/mg Cr (E1C) and 13 and 63 ng/mg Cr (PdG) (Figure 5J and Figure 6A,B, before Day 0). After the ovariectomy, followed by immediate implantation of encapsulated allogeneic ovarian tissue, the levels of E1C and PdG declined to 8–25 and 11–22 ng/mg Cr, respectively, in Primate B (Figure 6B). E1C increased between 130 and 200 days post-implantation, reaching 34 and 36 ng/mg Cr in the first and second rounds of implantation, respectively, with decline and peak occurring approximately every 7 days. PdG also increased, reaching 55 ng/mg Cr, similar to the measured levels of urinary E1C and PdG prior to ovariectomy. After 160 days, the encapsulated autografts from Primate A were explanted and replaced with encapsulated allografts from Primate C (Figure 6A). The levels of circulating E1C and PdG initially declined to 10 and 28 ng/mg Cr for E1C and 7 to 37 ng/mL Cr for PdG. After implantation of encapsulated ovarian allografts, the levels of E1C and PdG continued to increase, reaching 60 ng/mg Cr (E1C) at day 229 and 50 ng/mL Cr (PdG) at day 216 post-ovariectomy. Between the distinct peaks, the ovarian hormone levels appeared to fluctuate at lower values.

### 3.6. Ovarian Tissue Encapsulated in PEG Capsules Did Not Elicit an Immune Response

To assess whether the rhesus monkeys mounted an immune response against the encapsulated tissue, mixed lymphocyte culture was performed at the conclusion of the second round of implantations for all three animals. Peripheral blood mononuclear cells (PBMCs) from each animal were tested for reactivity to autologous cells (negative control), to allogeneic cells to which they were naïve (exposure of A to B, B to A, C to A, and C to B cells), or to cells from the ovarian allograft donor (exposure of A to C or B to C cells). We observed no significant proliferation of CD4+ T cells in response to allogeneic donors when compared to proliferation in response to autologous or third-party cells (negative controls) (Figure 6C) (*p* = 0.1889). Similarly, there was no significant spike in dividing CD8+ T cells in any of the animals when implanted with encapsulated allogeneic or autologous ovarian tissue (*p* = 0.8371). This finding was comparable to the observations after mixing responder cells with irrelevant, non-donor stimulators (Figure 6D).

## 4. Discussion

In patients experiencing POI, allogeneic cell-based therapy has the potential to restore hormonal function in a physiologically pulsatile and self-regulating manner, potentially avoiding the side effects of pharmacological HRT as well as the risk of cancer recurrence associated with auto-transplantation. Additional populations that could benefit from encapsulated ovarian allo- or auto-transplantation include women with genetic causes of POI or women with autoimmune diseases such as lupus that receive cyclophosphamide immunosuppressive therapy. Here, we demonstrated that an immune-isolating dual-layered PEG capsule prevents sensitization and sustains prolonged function in previously sensitized hosts, which is paramount to allowing treatment with sequential allografts for children or young adults with POI that will require treatment for decades.

In the past 5 years, we designed an immune-isolating capsule able to support the physiological function of implanted murine ovarian tissue and protect the allograft from immune rejection. Our work was built on the hypothesis that non-vascularized allogeneic ovarian tissue encapsulated in a hydrogel-based capsule responds to circulating stimuli and releases ovarian hormones, reaching systemic circulation by diffusion, subsequently restoring cyclic ovarian function in ovariectomized mice. To this end, we (1) engineered and characterized a novel dual-layered capsule composed of a poly(ethylene-glycol) (PEG) hydrogel, with a proteolytically degradable core and non-degradable shell; (2) demonstrated that encapsulated and implanted murine ovarian tissue restored estrous cyclicity and normalized levels of circulating follicle stimulating hormone for over 60 days; (3) demonstrated minimal to no inflammatory foreign body reaction around the capsule in both syngeneic and allogeneic models; (4) established that encapsulated ovarian allografts did not ”sensitize” a naïve host; and (5) showed that capsules protected the allografts implanted in previously sensitized hosts allowing repeated implantations without loss of function due to alloimmunity [35,36,37,38,40]. To test our mouse model in a more clinically relevant setting, we sensitized naïve mice by implanting non-encapsulated ovarian allografts, mimicking human patients who have been exposed to alloantigens. We demonstrated that encapsulated ovarian allografts were protected against rejection even in sensitized recipients.

The obvious differences between rodent and primate ovarian biology, such as the size, distribution, and number of ovarian follicles, the longer duration of folliculogenesis, and animal size and pubertal development, justified a pilot study with a small number of adolescent rhesus monkeys [45]. As a next step, we investigated the ability of an immunoisolating capsule to support follicular development and restore basal ovarian endocrine function in young ovariectomized rhesus monkeys implanted with autologous or allogeneic ovarian tissue. Importantly, to mimic the clinical scenario of puberty induction in young cancer survivors, the three animals in this study were young, peripubertal females that did not yet exhibit regular follicular and luteal phases of the menstrual cycle. Female rhesus macaques, similar to humans, have a long and complex period of maturation during adolescence, when they undergo a growth spurt and the development of secondary sex characteristics, followed by menarche, intense bone mineralization and height growth, epiphyseal closure, and finally, the onset of stable ovulatory menstrual cycles. These differences further support the use of rhesus monkeys to fully understand the late negative effects of non-physiological puberty induction on the same time scale as humans. Once the monkeys were ovariectomized and implanted with encapsulated ovarian tissue, they maintained basal levels of E1C and PdG throughout the entirety of the implantation period, similar to the hormone levels detected prior to ovariectomy. The animals were still too young to have regular ovarian cycles similar to those of mature females, but the presence of basal ovarian hormones following the implantations reflected follicular activation, follicle recruitment into the growing pool, and progression to the small antral stage, particularly in the autograft implant [44,45]. The histological analysis of the retrieved ovarian grafts revealed healthy follicles of all follicular stages (up to the antral stage), indicating that folliculogenesis was supported in the capsules. The presence of small antral follicles suggests that rhesus monkey ovarian tissue can undergo the necessary volumetric expansion during folliculogenesis to the small antral stage without capsule restrictions, most likely due to the degradation of the core and the viscoelastic properties of the shell. Upon visual evaluation after explant, there was no evidence of fibrous tissue or inflammation at the site, similar to our prior findings in rodents.

To determine whether implanted young rhesus monkeys mounted an immune response against the encapsulated allografts, mixed lymphocyte cultures were performed. We observed that when PBMCs from the animals were exposed to cells from the donor, there was no significant CD4+ and CD8+ T cell proliferation when compared to autologous or third-party controls. This finding suggests that the animals were not sensitized to the alloantigens and that the capsules isolated the ovarian allografts from the immune system of the recipient. Importantly, the PEG capsules were able to support folliculogenesis without compromising their immunoisolating capability.

We believe this report is the first to demonstrate the use of an immunoisolating capsule to support nonhuman primate folliculogenesis in a non-vascularized ovarian tissue graft while protecting the ovarian allograft from the immune system and maintaining basal levels of ovarian endocrine function. Whether restoration of cyclic ovarian function typical of adult female rhesus monkeys during regular menstrual cycles can be supported by the immunoisolating capsules remains to be determined. In summary, we have established that encapsulating allogeneic ovarian tissue in a PEG-based capsule protects the allograft, restores and maintains some ovarian endocrine function, prevents the sensitization of the host immune system, and functions similarly in sensitized and naïve mice as well as rhesus monkeys. Further studies assessing the ability of the dual-layered PEG capsule to support normal ovarian cyclicity, including corpus luteum formation, will be important to verify that the immunoisolation construct is functional. Long-term studies in monkeys can also inform on long-term efficiency, graft longevity, and potential safety concerns. Capsule modifications to support larger graft fragments and to promote vasculature formation around the capsule and greater diffusion of essential metabolites and nutrients without the risk of immune rejection may be essential. Lastly, in the future, this capsule may promote folliculogenesis and maturation of fertilizable oocytes in the encapsulated ovarian autograft, minimize the risk of cancer cells escaping the graft, and result in fully grown and mature eggs that can be retrieved and fertilized with the goal of restoring fertility to the patient.

## Figures and Tables

**Figure 1 bioengineering-10-00550-f001:**
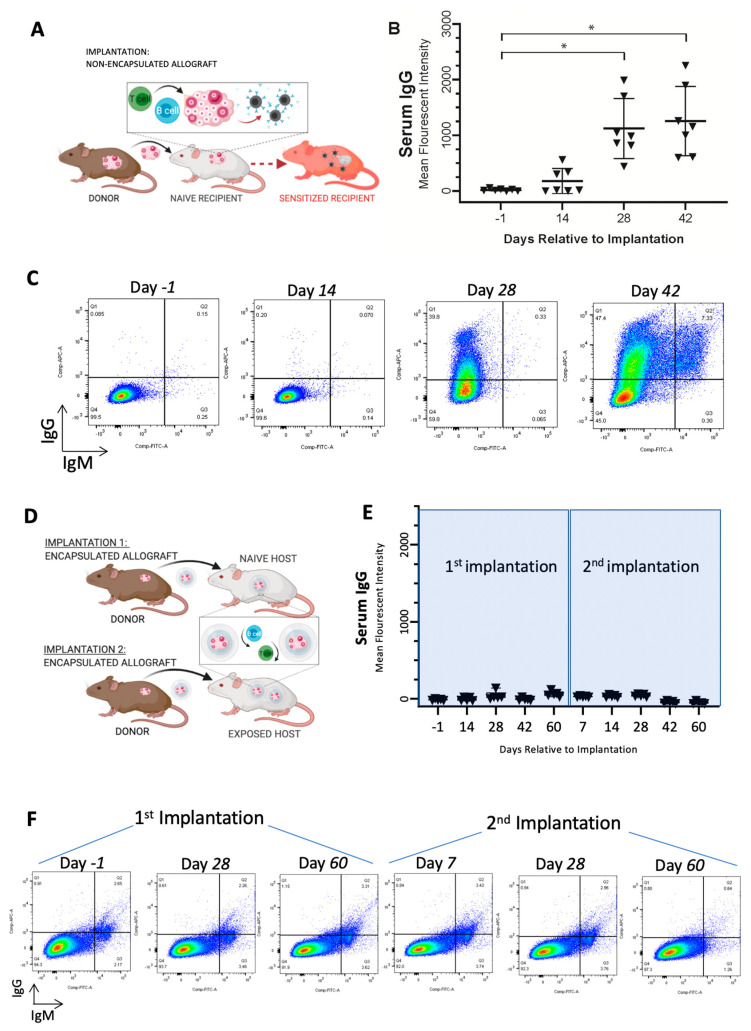
Implantation of encapsulated and non-encapsulated ovarian allografts in immune competent mice. (**A**) Schematic for implantation of non-encapsulated ovarian allograft, which causes sensitization of the recipients, production of allo-antibodies, T cell infiltration, and rejection. (**B**) Flow cytometric analysis of serum obtained from recipient mice with non-encapsulated ovarian allografts indicated average mean fluorescence intensity (MFI) ± SD of allo-specific IgG over time after implantation (*n* = 7 mice/timepoint). * statistical difference (*p* < 0.05) (**C**) Representative flow cytometry plots of binding of serum allo-specific antibodies from recipients of non-encapsulated ovarian allograft after implantation for up to 42 days. (**D**) Schematic for implantation of encapsulated ovarian allograft, which blocks interaction between the graft and recipient’s immune system and prevents sensitization of the recipient. (**E**) Average MFI ± SD of allo-specific IgG over time, obtained from serum of recipient mice by flow cytometry, following implantation of two consecutive encapsulated ovarian allografts (*n* = 5 mice/timepoint). (**F**) Representative flow cytometry plots detecting serum allo-specific antibodies in recipients implanted with two consecutive encapsulated ovarian allografts.

**Figure 2 bioengineering-10-00550-f002:**
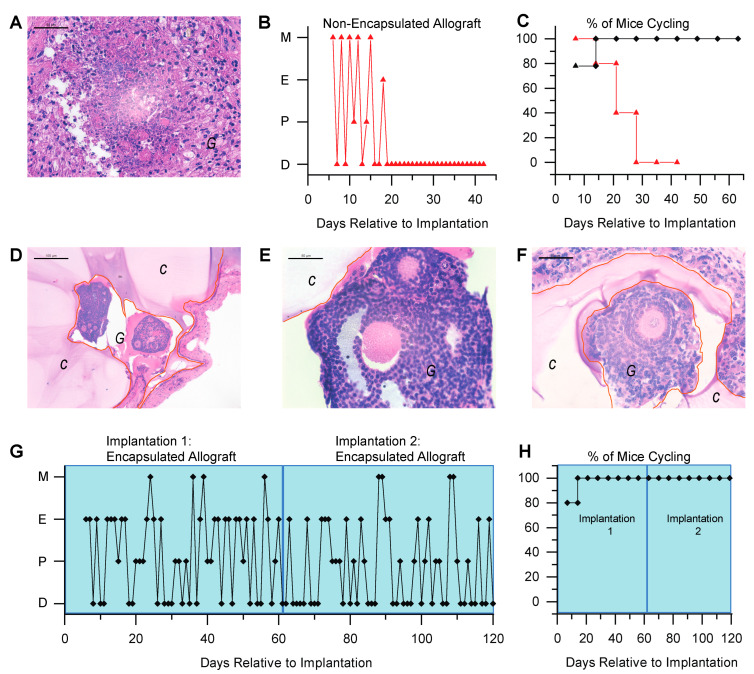
Encapsulation of ovarian allografts preserves the structure and function of ovarian tissue. (**A**) Representative histology of implanted non-encapsulated ovarian allograft. No phenotypically intact follicles were identified 6 weeks post-implantation. (**B**) Representative plot of estrous activity of mice receiving a non-encapsulated ovarian allograft; Metaestrus-M, Estrus-E, Proestrus-P, Diestrus-D. (**C**) Estrous cyclicity in mice implanted with non-encapsulated (red, *n* = 7) and encapsulated (black, *n* = 5) ovarian allografts. (**D**–**F**) Ovarian allografts retrieved after the first (**D**,**E**) and second round (**F**) of implants were completely surrounded by the hydrogel capsule and isolated from the host. Multiple follicles at various developmental stages (primordial through antral) were present. (**G**) Representative plot of estrous activity in mice receiving two rounds of encapsulated ovarian allografts. (**H**) Continued estrous cyclicity following two rounds of encapsulated ovarian allografts (*n* = 5). In (**A**,**D**–**F**), the capsule is denoted by ‘C’ and outlined in orange and graft is denoted by ‘G’. Scale bars represent 100 μm in (**D**) and 50 μm in (**A**,**E**,**F**).

**Figure 3 bioengineering-10-00550-f003:**
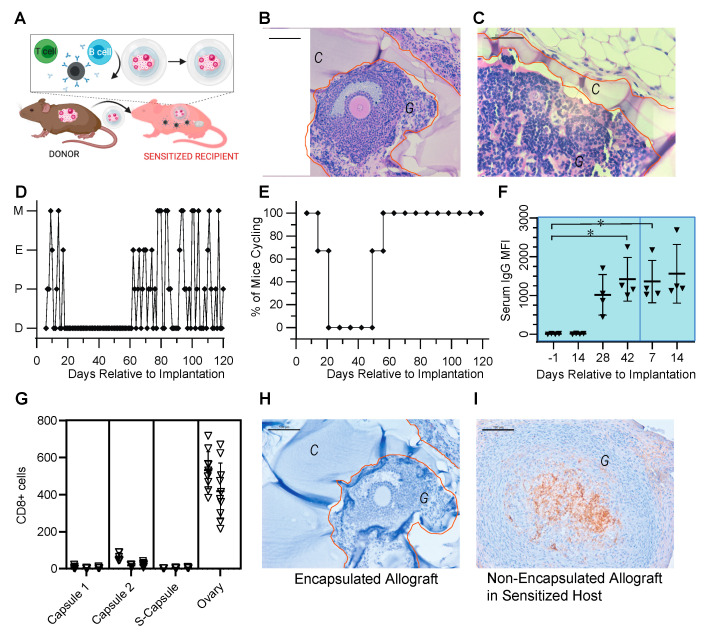
Encapsulated allografts restored ovarian endocrine function and were protected from rejection in sensitized hosts. (**A**) Schematic for implantation of non-encapsulated ovarian allograft with production of allo-antibodies, T cell infiltration, and rejection, followed by implantation of an encapsulated allograft. (**B**,**C**) Representative histology of encapsulated ovarian allografts retrieved after implantation demonstrating that the tissues were surrounded by the hydrogel capsule and isolated from the host. Multiple follicles at various developmental stages, antral (**B**) and preantral (**C**) were present. (**D**) Representative plot of estrous activity of mice receiving a non-encapsulated followed by an encapsulated ovarian allograft. (**E**) Summarized estrous cyclicity for all the mice implanted with non-encapsulated followed by encapsulated ovarian allografts (*n* = 3). Metaestrus-M, Estrus-E, Proestrus-P, Diestrus-D. (**F**) Average MFI ± SDI of allo-specific IgG in mice after implantation of two consecutive non-encapsulated ovarian allografts (*n* = 4). (**G**) CD8+ T cells, identified via IHC and reported as mean ± SD, were present in the retrieved encapsulated allografts after first (Capsule 1) and second implantations (Capsule 2), (*n* = 3 mice), in retrieved encapsulated allografts implanted in sensitized mice (S-Capsule), (*n* = 3 mice), and in retrieved non-encapsulated allografts implanted in a host (Ovary), (*n* = 2 mice) (* *p* <0.0001). Five to ten identically sized regions of interest (ROI) within the allografts were evaluated per animal. Each datapoint represents a cell count per ROI. Vertical clusters of datapoints represent counts within an ROI per animal. (**H**) Representative anti-CD8 IHC staining of encapsulated allografts for all encapsulated groups, (**I**). Representative anti-CD8 IHC staining of the non-encapsulated allograft. In (**B**,**C**,**G**,**H**), the capsule is denoted by ‘C’ and outlined in orange and graft is denoted by ‘G’.Scale bars represent 100 μm in (**B**,**H**,**I**) and 50 μm in (**C**).

**Figure 4 bioengineering-10-00550-f004:**
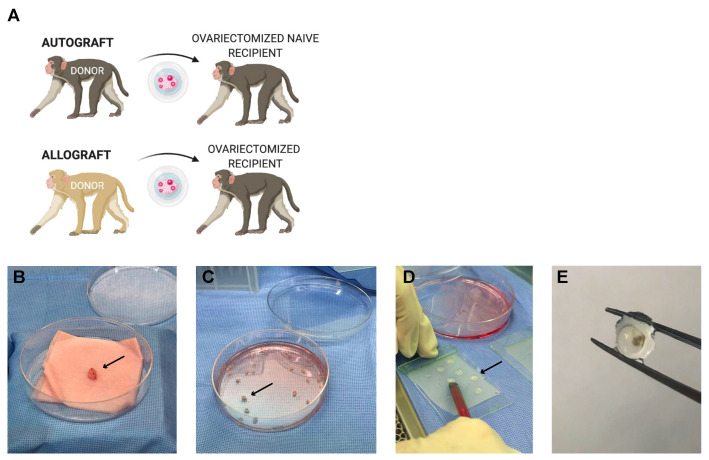
Harvest, processing, encapsulation, and explant of rhesus monkey ovarian tissue in immuno-isolating capsules. (**A**) Schematic for implantation of encapsulated auto- and allogeneic nonhuman primate ovarian tissue. (**B**) The ovary (arrow) was surgically removed from the donor animal and processed under sterile conditions. (**C**) The ovary was dissected into 1 mm^3^ (arrow) fragments and (**D**) encapsulated in immuno-isolating capsules (arrow) consisting of 28 uL of PEG components. (**E**) Capsules were retrieved post-implantation after 4 or 5 months with no visible blood vessel infiltration or fibrosis surrounding the capsules. The ovarian tissue was visible in the center of the capsule.

**Figure 5 bioengineering-10-00550-f005:**
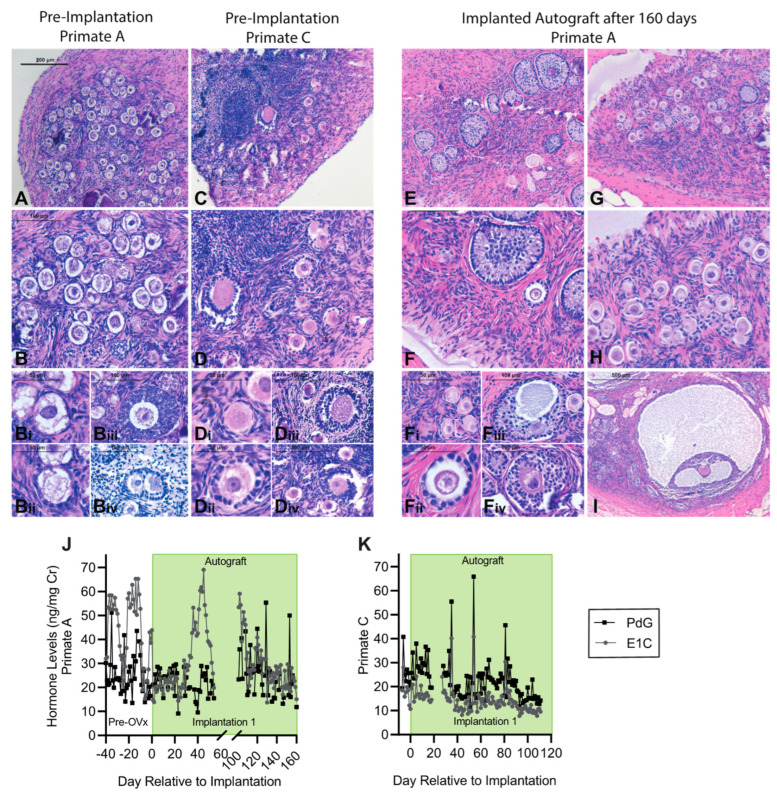
Histologic analysis of rhesus monkey donor ovarian tissue at implantation for Primate A (**A**,**B**,**Bi–iv**) and Primate C (**C**,**D**,**Di–iv**); autologous rhesus ovarian tissue encapsulated in immuno-isolating capsules for 5 months (**E**,**F**,**Fi–iv**,**G**–**I**). Scale bars: 200 μm (**A**,**C**,**E**,**G**), 100 μm (**B**,**D**,**F**,**H**), 50 μm (**Bi**,**Di**,**Fi**,**Bii**,**Dii**,**Fii**), 100 μm (**Biii**,**Diii**,**Fiii**,**Biv**,**Div**,**Fiv**), 500 μm (**I**). Urinary estrone conjugate (E1C) and pregnanediol 3-glucuronide (PdG) levels in animals that received (**J**) encapsulated autologous tissue for 5 months, and (**K**) encapsulated autologous tissue for 4 months. Levels are normalized to urinary creatinine (Cr) [48].

**Figure 6 bioengineering-10-00550-f006:**
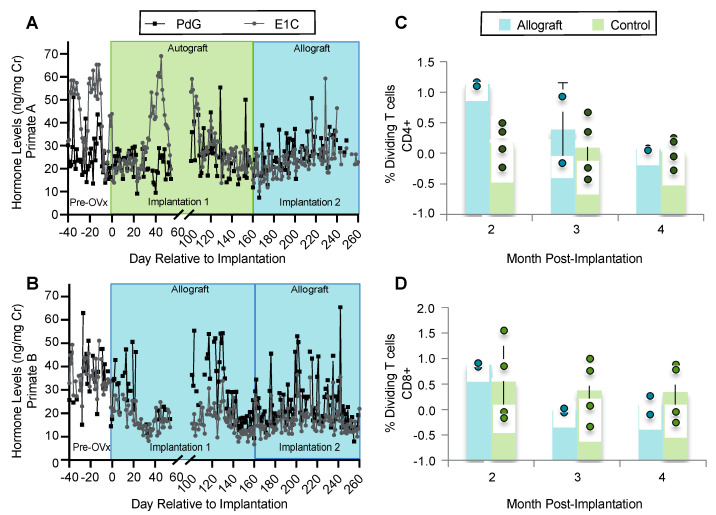
Urinary E1C and PdG levels and mixed lymphocyte culture in rhesus monkeys. Urinary E1C and PdG levels in rhesus monkeys that received (**A**) encapsulated autologous tissue for 5 months followed by encapsulated allografts for 4 months, (**B**) encapsulated ovarian allografts for 5 months during the first implantation followed with a second implantation of encapsulated allografts for 4 months. Levels are normalized to urinary creatinine. Mixed lymphocyte culture of recipient cells reacted with donor cells quantifying dividing (**C**) CD4^+^ and (**D**) CD8^+^ T cells. Bar graphs represent mean ± SD.

## Data Availability

The main data supporting the results of this study are available within the paper. The data used to produce the figures are available from Figshare with the identifier https://doi.org/10.6084/m9.figshare.c.5451663. The raw and analyzed datasets generated during the study related to flow cytometry are available for research purposes from the corresponding author upon reasonable request.

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
