# Peer review of "Encapsulated Allografts Preclude Host Sensitization and Promote Ovarian Endocrine Function in Ovariectomized Young Rhesus Monkeys and Sensitized Mice"

_bioengineering, 2023, doi:10.3390/bioengineering10050550_

Round 1

Reviewer 1 Report

There are some written lines copied from elsewhere desscribing how to write a scientific article. See last 9 lines of the introduction moreover the discussion shuold include concern if any about long term safety of this methods in the application to primates

Author Response

Comments and Suggestions for Authors

  1. There are some written lines copied from elsewhere describing how to write a scientific article. - We apologize for the oversight. This must have happened when we copied the text of the manuscript into the template provided by the journal. The aforementioned text was removed.
  2. See last 9 lines of the introduction moreover the discussion should include concern if any about long term safety of this methods in the application to primates

Thank you for this comment. We added the following sentence in the discussion:

Long-term studies in monkeys can also inform on long-term efficiency, graft longevity and potential safety concerns.

Reviewer 2 Report

1.     The work targeted a clinical issue by designing an innovative solution to help a specific population of female cancer patients who are suffering from toxic effects of essential treatment, the solution not only solves the immaturity of the ovary only but protect the immature girls from other compinged complication that caused by immature ovaries and hormonal shortage such as incomplete bone growth and close, fat disturbance, and cardiovascular issue and other functions related to gonad hormones.  The work is deserved to be translated to clinical study with fund to save hundreds of girls suffer from this case.

2.     One of the important praises, that researchers have experienced on two mammalian models rodents and Primate which acquired the work confirmation and validation to translate it to clinical trials  

3.     Recommendations:

·      Along term experiments and observation are needed to explore and detect any unexpected side effects.  

·      Testosterone conversion rates and levels should be estimated

·      Other cautions for adaptations by the body to allow infusion/ filtration of immune cells via capsule of the transplanted ovary

·      Investigate lymph circulation which is the primary way for migration of cancer cells to the graft as well as lymphocytes   

Introduction

4.     Line 110-118: It seems that this section is added by error. It should be deleted from the introduction

5.     Authors have to add at the end of the introduction aim and what is the gap would be covered by this work, age of targeted patients/animal  

Methodology

6.     Line 131: Author used adult female mice to induce POI while the target was immature cases, even if it is pilot study  

7.     Hydrogel preparation: Author have to demonstrate it in the manuscript by inserting real lab photos or at least by graphics demonstration.

8.     Line 167: name the devise of imaging

9.     Line 254: E1C and PdG: shall be identified by their full name for the first time of mention

Subcutaneous implantation:

10.  Line 168: Did researchers implant the encapsulated and non-capsulated ovaries at the same time?

11.  Line 181: serum for titer measurements for antibody (Ab) was done before and after implantation, author have to identify what was the day post-surgery that they started AB measurements as the release of IgM starts after latent periods from exposure to (foreign body=antigen) which is in this case (non-capsulated ovary). Additionally, IgM remains high two weeks after immunization, then it decreases, then IgG increases markedly

Please clarify this point in days to avoid any overlap by these two Abs.

12.  Authors have to state the objectives of repeated implantations

13.  Lines 241 & 248: Why was the tissue fragmented to 60 fragments… and why did researcher use these numbers of fragments? Please clarify

Discussion

14.  Researchers have to design graph or write section on the mechanism between blood circulation/ nutrients cross between double encapsulated implanted ovaries and body circulation. As well as how did prevent filtration of cellular immunity, and other humoral immune materials such as interferon, TNF, histamine, etc….Although the implantation maintains the endocrine functions, please explain carefully these results  

If this model succeeded, it may help in other organ transplantation and avoid immune rejection

15.  Line 535: correct “suggest” to “suggests” and other grammatical error in the manuscript

Authors that they succussed in isolating ovary from immune system, which is used as target, but recent studies confirmed that immune system is needed for menopause phase to stop folliculogenesis, So, author to state limitation of the work that they didn’t use senile model to test the transplantation for the long term of the receiving murine and primate

Ref: Al-Suhaimi, E.A., Khan, F.A., Homeida, A.M. (2022). Regulation of Male and Female Reproductive Functions. In: Al-Suhaimi, E.A. (eds) Emerging Concepts in Endocrine Structure and Functions. Springer, Singapore. https://doi.org/10.1007/978-981-16-9016-7_9

References

I surprised that the work missed to cite recent references. There is a gap of 4 years which was not covered by related references during 2020-2023 which hold the work away from any update.

I suggest these reference:

1.     ACS Nano 2022, 16, 4, 5246–5257 Publication Date:March 16, 2022

https://doi.org/10.1021/acsnano.1c07237

2.     Curr. Oncol. 2022, 29(3), 1583-1593; https://doi.org/10.3390/curroncol29030133

3.     Cancers 2022, 14(6), 1585; https://doi.org/10.3390/cancers14061585

4.     Al-Suhaimi, E.A., Khan, F.A., Homeida, A.M. (2022). Springer https://doi.org/10.1007/978-981-16-9016-7_9

  1. https://doi.org/10.1080/10520295.2022.2075566. 2023

Author Response

Comments and Suggestions for Authors

  1. The work targeted a clinical issue by designing an innovative solution to help a specific population of female cancer patients who are suffering from toxic effects of essential treatment, the solution not only solves the immaturity of the ovary only but protect the immature girls from other compinged complication that caused by immature ovaries and hormonal shortage such as incomplete bone growth and close, fat disturbance, and cardiovascular issue and other functions related to gonad hormones.  The work is deserved to be translated to clinical study with fund to save hundreds of girls suffer from this case.

Thank you for the kind and positive feedback!

  1. One of the important praises, that researchers have experienced on two mammalian models rodents and Primate which acquired the work confirmation and validation to translate it to clinical trials  - we appreciate the reviewer mentioning the translational nature of our work and the validation performed in two different animal models of premature ovarian insufficiency.
  2. 3.Recommendations:
  • A long term experiments and observation are needed to explore and detect any unexpected side effects.  – Thank you for tis suggestion! We absolutely agree that long-term biocompatibility, safety and efficiency studies have to be performed in non-human primates to argue that the described technology is a safe treatment for premature ovarian insufficiency. Our goal in the current study was to test whether repeated implantations in a mouse would cause sensitization, augmented immune response and faster rejection of the graft and whether circulating antibodies in a previously sensitized mouse would cause a stronger immune response and shorten the function of the graft. The studies in mice were performed over 4 months. The purpose of the described studies in the non-human primates was a proof of concept and a first evidence that encapsulated and immune isolated monkey allografts can survive for months. With the promising results we obtained here we will be pursuing funding to perform the suggested long-term experiments.

We added a sentence addressing the importance of long-term studies in the discussion:

Long-term studies in monkeys can also inform on long-term efficiency, graft longevity and potential safety concerns.

  • Testosterone conversion rates and levels should be estimated – In this study we focused only on the levels of E2 and P in the urine collected from the monkeys. In mice the blood volume that can be collected weekly is low and presents a significant limitation to what can be measured As a result we diverge to vaginal cytology to establish whether mice are experiencing ovarian function. The focus of the study in mice was on immune responses in naïve and sensitized animals and the small amounts of serum was used to test the presence of circulating allo-antibodies. In earlier publications we demonstrated that the implanted ovarian allograft decreased the levels of circulating FSH in ovariectomized mice which also correlated with estrous cyclicity determined using vaginal cytology. We agree with the reviewer’s suggestion and will include the testosterone conversion rates analysis in our future studies with monkeys.
  • Other cautions for adaptations by the body to allow infusion/ filtration of immune cells via capsule of the transplanted ovary – We agree with this comment. The importance of balancing between sufficient diffusion and preventing infiltration of immune cells is the foundation of this technology. IHC of the encapsulated grafts for the presence of CD8 and CD4 positive cells confirmed that the grafts were protected while non-encapsulated grafts had strong positive staining. Diffusion of soluble factors and small molecules through these hydrogels was extensively studies and reported elsewhere.

  • Investigate lymph circulation which is the primary way for migration of cancer cells to the graft as well as lymphocytes  - We agree that a closer look into lymph nodes would further validate the lack of immune response in animals implanted with encapsulated allografts. Location and isolation of the draining lymph nodes proved to be difficult in mice, but will be definitely included in a larger monkey study.  

Introduction

  1. Line 110-118: It seems that this section is added by error. It should be deleted from the introduction – We apologize for the oversight. This must have happened when we copied the text of the manuscript into the template provided by the journal. The aforementioned text was removed.
  2. Authors have to add at the end of the introduction aim and what is the gap would be covered by this work, age of targeted patients/animal  - We included the aim of the study and focused the last paragraph of the introduction on the translation of this technology

Methodology

  1. Line 131: Author used adult female mice to induce POI while the target was immature cases, even if it is pilot study – This is a valid point and we agree with the comment. The aim of this work was to investigate the function of the encapsulated allograft for which we believed that ovariectomized 6-8 weeks old female mice served as a suitable model. Our future studies will include prepubertal mice (although ovariectomies in prepubertal, 3-4 weeks old mice proved to be ethically and technically challenging) and large animal models.
  2. Hydrogel preparation: Author have to demonstrate it in the manuscript by inserting real lab photos or at least by graphics demonstration. – Thank you for this comment. We added citations of the earlier papers where the hydrogel preparation was described in depth and we added a supplemental figure (SF 1). The macroscopic real photo of the process for preparation of the immune-isolating capsule is also included as Figure 5 B-E.
  3. Line 167: name the devise of imaging- Thank you for this comment. We added: “Leica M60 stereo microscope”
  4. Line 254: E1C and PdG: shall be identified by their full name for the first time of mention – thank you for this comment. We edited the subtitle on page 7 to state:

Primate urinary estrone conjugate (E1C) and pregnanediol glucuronide (PdG) analysis

Subcutaneous implantation:

  1. Line 168: Did researchers implant the encapsulated and non-capsulated ovaries at the same time? – Thank you for this comment. We first implanted non-encapsulated allografts to activate the immune response, induce rejection of the tissue and sensitize the mice. This was confirmed by the presence of circulating allo-antibodies. In the second step of the study (60 days post implantation of the non-encapsulated allograft) we implanted the encapsulated allograft to test whether the capsule can protect the allograft in the presence of circulating allo-antibodies. This text can be found lines 157-169

“BALB/c mice were implanted subcutaneously with a non-encapsulated ovary from 6-8 days old CBA x C57BL/6 (F1) mice for 42 days (Figure 1A) (n=7) followed by a subsequent implantation of another non-encapsulated F1 ovary for 14 days in a subset of these mice (Figure 3F) (n=4 mice) For experimental groups, BALB/c mice were (i) implanted with encapsulated ovarian allogeneic (F1) tissue for 60 days followed by a subsequent implantation of encapsulated ovarian allografts for another 60 days (n=5 mice) or (ii) implanted with non-encapsulated F1 ovarian tissue for 60 days to induce sensitization followed by a subsequent implantation of encapsulated ovarian allografts for 60 days (n=3 mice). A total of 15 mice were used in these studies. Each mouse was implanted with two replicate grafts or non-encapsulated grafts.”

  1. Line 181: serum for titer measurements for antibody (Ab) was done before and after implantation, author have to identify what was the day post-surgery that they started AB measurements as the release of IgM starts after latent periods from exposure to (foreign body=antigen) which is in this case (non-capsulated ovary). Additionally, IgM remains high two weeks after immunization, then it decreases, then IgG increases markedly

Please clarify this point in days to avoid any overlap by these two Abs. – Thank you for this insightful comment. We added the statement that Ab measurements were done one day before the implantation followed by bi-weekly measurements: -1, 14, 28, 42, 56 (7 days post second implant), 70 (21 days post second implant). This was highlighted in the text on page 9.

  1. Authors have to state the objectives of repeated implantations - Thank you for this comment. We stated the objectives of repeated implantations in the introduction and the results.

Sensitization of the host and presence of circulating allo antibodies may shorten graft longevity and elicit local and systemic adverse effects. Here, our objective was to demonstrate that encapsulation of ovarian allograft tissue prevented sensitization of the host that would allow repeated implantations of encapsulated allografts without the risk of rejection. As expected, implantation of non-encapsulated allogeneic ovarian tissue elicited an immune response mediated by T and B cells, resulting in sensitization of the host against the donor with a 60-fold increase in allo-specific IgG and cell-mediated destruction of the graft tissue (Figure 1A – study design, 1B, C). Allo-specific IgG antibodies significantly increased from undetectable post-implantation to an average of 1123 mean fluorescence intensity (MFI) on day 28 (*, p=0.0036) and to an average of 1256 MFI on day 42 (*, p=0.0046) (n=7), confirming allogeneic immune reaction between the two strains of mice. We then assessed if implantation of encapsulated allografts allowed repeat implantation without the risk of rejection (Figure 1D – study design). To block immune-mediated graft damage, we created a dual-layered immuno-isolating capsule that creates a barrier between the allograft and the host to minimize the host exposure to allo-antigens and prevent stimulation of allo-immunity and graft damage owing to pre-existent allo-immunity. In contrast to non-encapsulated controls, encapsulated ovarian allografts did not sensitize recipients, which was confirmed by undetectable levels of circulating allo-antibodies. Following the first and subsequent implantations of encapsulated ovarian allografts (Figure 1D) the levels of circulating allo-specific IgGs remained at the pre-implant level, ranging from non-detectable to 138 MFI for up to 60 days post-implantation (Figures 1E and F). It should be noted that there was a statistically significant difference between IgG MFI values pre-implantation (-1 day timepoint) and 60 days after the first round implantation as well as 7 and 28 days after the second round implantation; however, the small difference in IgG MFI values detected is not biologically significant. We concluded that neither primary nor secondary implantation of encapsulated allografts sensitized recipients because the levels of circulating IgG allo-antibodies did not increase after the implantations of encapsulated allografts.

  1. Lines 241 & 248: Why was the tissue fragmented to 60 fragments… and why did researcher use these numbers of fragments? Please clarify - Thank you for this comment. We edited the text for more clarification:

The primate ovaries were dissected into cubes measuring 1x1x1mm (1 mm3) contributing to approximately 60 fragments, which were transferred into maintenance media

Discussion

  1. Researchers have to design graph or write section on the mechanism between blood circulation/ nutrients cross between double encapsulated implanted ovaries and body circulation. As well as how did prevent filtration of cellular immunity, and other humoral immune materials such as interferon, TNF, histamine, etc….Although the implantation maintains the endocrine functions, please explain carefully these results  

If this model succeeded, it may help in other organ transplantation and avoid immune rejection

Thank you for the suggestions and referring to this technology potentially applicable to other systems. We prefer not to make any claims that cannot be supported by data and application to other systems would be an overreach at this point.

We and others have demonstrated that hydrogels such as poly(ethylene glycol) and alginate are nanoporous and do not allow infiltration of the cells, which are significantly larger than the pores. With regards to diffusion of nutrients we and others have demonstrated that small molecules and gases undergo exchange via diffusion while larger molecules such as antibodies are slower to penetrate. We also performed specific staining of the hydrogels to test for the presence of T cells, which proved negative.

  1. Line 535: correct “suggest” to “suggests” and other grammatical error in the manuscript

Authors that they succussed in isolating ovary from immune system, which is used as target, but recent studies confirmed that immune system is needed for menopause phase to stop folliculogenesis, So, author to state limitation of the work that they didn’t use senile model to test the transplantation for the long term of the receiving murine and primate

Ref: Al-Suhaimi, E.A., Khan, F.A., Homeida, A.M. (2022). Regulation of Male and Female Reproductive Functions. In: Al-Suhaimi, E.A. (eds) Emerging Concepts in Endocrine Structure and Functions. Springer, Singapore. https://doi.org/10.1007/978-981-16-9016-7_9

References

I surprised that the work missed to cite recent references. There is a gap of 4 years which was not covered by related references during 2020-2023 which hold the work away from any update.

I suggest these reference:

  1. ACS Nano2022, 16, 4, 5246–5257 Publication Date:March 16, 2022

https://doi.org/10.1021/acsnano.1c07237

  1. Curr. Oncol.202229(3), 1583-1593; https://doi.org/10.3390/curroncol29030133
  2. Cancers202214(6), 1585; https://doi.org/10.3390/cancers14061585

  1. Al-Suhaimi, E.A., Khan, F.A., Homeida, A.M. (2022). Springerhttps://doi.org/10.1007/978-981-16-9016-7_9

  1. https://doi.org/10.1080/10520295.2022.2075566. 2023

We apologize for the oversight and missing the recent and important articles. We have updated our citation list and included the missing publications.

Reviewer 3 Report

The study is well-designed; however, some revisions could improve the quality of the manuscript. The manuscript should be revised in response to the comments.

- On page 3, lines 110-118, "The introduction should briefly place … document for further details on references." These sentences appear to be a reviewer's comment on the paper and should be removed.

- The main idea that runs throughout the study, as well as the aim of the study, should be included in the final paragraph of the "Introduction" section.

- The number of replicates for each test should be specified.

- On page 4, line 163, "et al." should be written in italics.

- Why did the authors include a subsection just for figures? Each figure should be placed next to the relevant text.

- Many of the statements in the "Discussion" section require citations.

- The references are not up-to-date. The authors should discuss the obtained results using the most recently published articles.

Author Response

The study is well-designed; however, some revisions could improve the quality of the manuscript. The manuscript should be revised in response to the comments.

Thank you for the kind and positive feedback!

- On page 3, lines 110-118, "The introduction should briefly place … document for further details on references." These sentences appear to be a reviewer's comment on the paper and should be removed- We apologize for the oversight. This must have happened when we copied the text of the manuscript into the template provided by the journal. The aforementioned text was removed.

- The main idea that runs throughout the study, as well as the aim of the study, should be included in the final paragraph of the "Introduction" section.

Thank you for this comment, we corrected this mistake.

- The number of replicates for each test should be specified. - we included the number of replicates, thank you!

- On page 4, line 163, "et al." should be written in italics. – Thank you for this comment. We adjusted the text

- Why did the authors include a subsection just for figures? Each figure should be placed next to the relevant text.

We have followed the template provided by the journal , which  unfortunately  does not allow placing figures next to the relevant text and doing so would make the figures too small and ineligible to view.

- Many of the statements in the "Discussion" section require citations.

- The references are not up-to-date. The authors should discuss the obtained results using the most recently published articles.

We apologize for the oversight and missing the recent and important articles. We have updated our citation list and included the missing publications.